# Imiquimod Is Effective in Reducing Cervical Intraepithelial Neoplasia: A Systematic Review and Meta-Analysis

**DOI:** 10.3390/cancers16081610

**Published:** 2024-04-22

**Authors:** Balázs Hamar, Brigitta Teutsch, Eszter Hoffmann, Péter Hegyi, Andrea Harnos, Péter Nyirády, Zsombor Hunka, Nándor Ács, Ferenc Bánhidy, Zsolt Melczer

**Affiliations:** 1Centre for Translational Medicine, Semmelweis University, 1088 Budapest, Hungary; teutschbrigitta@gmail.com (B.T.); h.eszter@icloud.com (E.H.); hegyi2009@gmail.com (P.H.); acsnandor@gmail.com (N.Á.); banhidyferenc@hotmail.com (F.B.); melczerzsolt@gmail.com (Z.M.); 2Department of Obstetrics and Gynecology, Semmelweis University, 1088 Budapest, Hungary; nyiradyp@gmail.com (P.N.); hunka.zsombi@gmail.com (Z.H.); 3Institute for Translational Medicine, Medical School, University of Pécs, 7621 Pécs, Hungary; 4Institute of Pancreatic Diseases, Semmelweis University, 1088 Budapest, Hungary; 5Department of Biostatistics, University of Veterinary Medicine, 1078 Budapest, Hungary; harnos.andrea@gmail.com; 6Department of Urology, Semmelweis University, 1082 Budapest, Hungary

**Keywords:** Imiquimod, gynecologic cancer risk reduction, HPV, cervical cancer

## Abstract

**Simple Summary:**

There are publications on the use of Imiquimod in cervical intraepithelial neoplasia (CIN) and HPV clearance; however, the literature is not consistent about its efficacy. Moreover, in cervical precancers, surgical solutions are widely accepted therapies, despite their association with increased obstetrical complications, such as miscarriage and preterm birth. Therefore, a conservative solution is needed. Topical Imiquimod reduced CIN and enhanced HPV clearance, though surgical intervention conization was found to be more effective than Imiquimod treatment. Side effects were common, though mostly mild. Topical Imiquimod could be a valuable therapeutic option for CIN patients, especially for women who have future pregnancy desires. Imiquimod should be incorporated into guidelines as evidence shows it is effective and safe.

**Abstract:**

Introduction: Topical Imiquimod is an immune response modifier approved for the off-label use of vulvar intraepithelial neoplasia. We conducted this systematic review and meta-analysis to investigate the efficacy and safety of Imiquimod in treating cervical intraepithelial neoplasia (CIN) and human papillomavirus (HPV)-positive patients. Methods: The study was prospectively registered (CRD420222870) and involved a comprehensive systematic search of five medical databases on 10 October 2022. We included articles that assessed the use of Imiquimod in cervical dysplasia and HPV-positive patients. Pooled proportions, risk ratios (RRs), and corresponding 95% confidence intervals (CIs) were calculated using a random effects model to generate summary estimates. Statistical heterogeneity was assessed using *I*^2^ tested by the Cochran Q tests. Results: Eight articles reported on 398 patients who received Imiquimod out of 672 patients. Among CIN-2–3 patients, we observed a pooled regression rate of 61% (CI: 0.46–0.75; *I*^2^: 77%). When compared, Imiquimod was inferior to conization (RR: 0.62; CI: 0.42–0.92; *I*^2^: 64%). The HPV clearance rate in women who completed Imiquimod treatment was 60% (CI: 0.31–0.81; *I*^2^: 57%). The majority of side effects reported were mild to moderate in severity. Conclusions: Our findings indicate that topical Imiquimod is safe and effective in reducing cervical intraepithelial neoplasia and promoting HPV clearance. However, it was found to be inferior compared to conization. Imiquimod could be considered a potential medication for high-grade CIN patients and should be incorporated into guidelines for treating cervical dysplasia.

## 1. Introduction

Cervical intraepithelial neoplasia (CIN) 2–3 is the precursor lesion of cervical cancer, one of the leading causes of cancer-related death in women. In 2020, there were 604,000 new cases and 342,000 deaths worldwide, according to GLOBOCAN [1]. High-risk human papillomavirus (HPV) is the predominant factor (99.7%) responsible for cervical cancer [2].

While only a minority of cases progress to invasive cancer after years of persistence, most patients experience regression of CIN to a normal condition [3]. For histologic high-grade intraepithelial lesions (HSILs), excisional treatment is preferred according to the 2019 American Society for Colposcopy and Cervical Pathology (ASCCP) Risk-Based Management Consensus Guideline [4]. However, these procedures may impact pregnancy outcomes, such as preterm delivery, premature rupture of membranes, and low birth weight [5,6]. Moreover, the persistence of HPV has been linked to an increased recurrence rate following surgical intervention [7]. Consequently, alternative conservative therapies are necessary to reduce the frequency of surgical interventions and associated complications.

Topical Imiquimod has been approved by the US Food and Drug Administration (FDA) for treating external genital and perianal warts, basal cell carcinoma, and actinic keratoses [8]. This compound is believed to activate immune cells as a Toll-like receptor-7 agonist. It exerts its antiviral effects by activating dendritic cells and inducing cytokines such as tumor necrosis factor-alpha (TNF-α), interferon-alpha (IFN-α), and interleukins (ILs) [9]. Multiple studies have shown that Imiquimod could be a potential conservative treatment for precursor cervical lesions by accelerating viral clearance [10,11,12]. However, some other publications have found that it is ineffective in reducing CIN [13]. There were no meta-analyses on the subject to answer this important question.

In this present study, based on the available literature, we aimed to determine the efficacy and safety of topical Imiquimod therapy in reducing the incidence of cervical intraepithelial neoplasia (CIN) and its impact on HPV clearance.

## 2. Materials and Methods

We followed the PRISMA 2020 (Appendix A) for conducting a systematic review and meta-analysis, as recommended by the Cochrane Handbook for Systematic Reviews of Interventions [14,15] (see Appendix A). The study design and protocol were registered in PROSPERO (CRD420222870), and we adhered to them completely.

### 2.1. Search Strategy

The complete search key is provided in the Appendix A. During the systematic search, the following search strategy was used: ‘Imiquimod’, ‘cervical intraepithelial neoplasia’, ‘cervical dysplasia’, ‘human papillomavirus’.

### 2.2. Literature Search and Eligibility Criteria

A systematic search was conducted using five major databases: MEDLINE (through PubMed), Embase, Cochrane Central Register of Controlled Trials (CENTRAL), Scopus, and Web of Science until 10 October 2022. No restrictions or filters were applied during the search. We used two frameworks to describe the eligibility criteria in the articles. First, the CoCoPop framework was used in studies with no comparators to assess. We investigated women with cervical intraepithelial neoplasia (Population) who applied topical Imiquimod (Context). We determined cervical dysplasia regression, estimation of treatment success, assessment of HPV clearance, and adverse events (Condition). Afterwards, the PICO framework was used. We assessed women (P) with cervical dysplasia or who were HPV positive. In the intervention group (I), women had to receive topical Imiquimod products for their cervical disease. Patients in the comparator group (C) received the standard treatment, predominantly surgical solutions: conization, cryotherapy, laser therapy or expectant management. Outcome (O) parameters included the assessment of cervical dysplasia regression, assessment of HPV clearance, and adverse events [16]. Cervical dysplasia regression was defined as the absence of dysplasia or regression from CIN 2–3 to CIN 1. HPV clearance was effective when the original HPV types could not be detected after treatment. Cohorts, case-control studies, and randomized controlled trials (RCT) were accepted. Only studies with patient follow-up were included. We imposed no language restrictions; non-English articles were translated into English and evaluated afterwards.

### 2.3. Selection Process and Data Collection

Article selection was performed using the reference management program EndNote X9. Duplicate removal was conducted by two independent reviewers (B.H., H.E.) at each stage: after title and abstract selection and during full-text selection. Cohen’s kappa coefficient (κ) was used to measure the level of agreement [17]. Disputed articles were resolved by a third independent reviewer (H.Z.S.).

Pre-defined variables were described in a Microsoft Excel spreadsheet (Windows 11 Pro 10) by two independent reviewers (B.H., E.H.). The following variables were extracted: first author, year of publication, digital object identifier, study type, study design, country, study period, centers, and duration of follow-up. The following were extracted for both the intervention group and the control group: patient numbers, patient age, pregnancy status, smoking status, number of sexual partners, histological findings (cervical intraepithelial neoplasia 2–3), and HPV status. The dose, duration, and application form were recorded in the intervention group. Outcomes were collected in two-by-two tables. Whenever possible, risk ratios (RRs) were extracted directly. Intention-to-treat (ITT) and per-protocol (PP) data were collected from RCTs. Response rate data were recorded separately when available. Adverse events were collected using the National Cancer Institute (NIH) website’s Common Terminology Criteria for Adverse Event protocols [18]. Adverse events were graded from 1 to 4 for the following: fatigue, headache, myalgia, flu-like symptoms, fever, abdominal pain, vaginal pruritus, vaginal discharge, vaginal bleeding, and inflammation. A third reviewer (Z.S.H.) resolved the conflict in case of disagreements.

### 2.4. Risk of Bias and Quality Assessment of Included Articles

Assessing bias and quality risk of bias and quality assessment depended on study type. RCTs were evaluated using the Risk of Bias II (ROB II) tool, while non-randomized interventions used the Risk of Bias in Non-Randomized Studies (ROBINS I) [19,20]. Response rates without a control group were assessed with the Joanna Briggs Institute (JBI) Critical Appraisal Checklists [21]. GRADE was applied to grade evidence, and a Summary of Findings Table was formulated using GradePro [22]. Two reviewers (B.H., E.H.) conducted assessments, with disputes resolved by a third reviewer (Z.S.H.).

### 2.5. Synthesis Methods

In data synthesis, both qualitative and quantitative analyses utilized R statistical programming language (R version 4.3). Quantitative synthesis required a minimum of three studies, presented in forest plots. Subgroup analyses were conducted based on article type and cervical dysplasia grade. RCTs’ ITT data were analyzed, while other study types were grouped as cohorts. For cervical dysplasia, subgroups included studies without CIN, CIN 1–2–3, and CIN 2–3. Per-protocol data were analyzed for RCTs with complete treatment. Risk ratios (RR) with 95% confidence intervals (CI) assessed effect sizes. The Clopper-Pearson method calculated CIs. Statistically significant results excluded the null value within pooled CI [23]. Forest plots summarized meta-analysis findings. Higgins & Thompson’s *I*^2^ assessed heterogeneity, with τ^2^ indicating variance [24].

Heterogeneity levels were categorized: 0–40% possibly not important, 30–60% moderate, 50–90% substantial, 75–100% considerable. Subgroups used a fixed effects “plural” model. Cochrane Q test evaluated subgroup differences [25]. To assess the difference between the subgroups, a “Cochrane Q” test was used between subgroups (Harrer et al. 2021). The null hypothesis was rejected on a 5% significance level [26].

Publication bias was not assessed due to limited studies (<10).

## 3. Results

### 3.1. Search and Selection

Our systematic search identified 3141 articles from five databases. After removing duplicates, 2218 articles were analyzed for title and abstract selection. In the full-text selection, 13 eligible articles were screened. Finally, we found eight eligible articles for quantitative and qualitative synthesis (see Figure 1).

### 3.2. Basic Characteristics of Included Studies

The eligible articles were published between 2012 and 2022. Regarding demographics, the mean age of women included in the studies was 30.41 years (±2.15). The mean follow-up time was 18.62 (±12.00) months. In six studies, women had histologically proven CIN 2–3; in the seventh study, both cytology and histology were used [27]. Quantitative synthesis was possible only for a subpopulation with HPV status. HPV tests were performed in seven articles. Details about the doses and application of Imiquimod can be found in Appendix A.

Altogether, 672 patients were included from the eight studies [10,11,12,13,27,28,29,30], with 398 women receiving Imiquimod treatment. Detailed baseline characteristics can be found in Table 1.

### 3.3. CIN 2–3 Regression

A total of 294 women received topical Imiquimod treatment for CIN 2–3 [10,12,27,28,30]. These patients showed a regression rate of 61% (CI: 0.46–0.75; *I*^2^: 77%) to CIN 1 or no disease after topical Imiquimod therapy (see Figure 2). The subgroup analysis based on study type revealed a histologic regression rate of 59% (CI: 0.47–0.70) in the ITT-RCTs, while the response rate in the cohort studies was 64% (CI: 0–1.00 *I*^2^: 94) (see Figure 2). In the PP population, which consisted of 155 patients, the regression rate was 67% (CI: 0.54–0.78; *I*^2^: 0%) (see Appendix A).

Two articles investigated the efficacy of topical Imiquimod [10,11]. In both studies, the RR for CIN regression was higher when Imiquimod was compared to no treatment (RR: 1.87; CI: 1.12–3.10 and RR: 2.37; CI: 1.25–4.48, respectively) (see Appendix A).

In the experimental group, 196 women received Imiquimod treatment, while 196 women were in the control group who underwent conization [10,12,28,30]. Women who underwent conization had a 38% decrease in the risk of persistence or progression in CIN compared to the women who applied Imiquimod (RR: 0.62; CI: 0.42–0.92; *I*^2^: 64%) (see Figure 3). The subgroup analysis showed a randomized clinical trial where conization was superior to Imiquimod and had a 44% decrease in the risk of unsuccessful treatment (RR: 0.56; CI: 0.43–0.74) (see Figure 3) [28]. Likewise, in the PP analysis, Imiquimod was not a superior intervention to conization (RR: 0.78; CI: 0.56–1.07; *I*^2^: 0%) (see Appendix A). 

### 3.4. Imiquimod on HPV Clearance

Among the 254 patients who received Imiquimod treatment, 50% (CI: 0.35–0.64; *I*^2^: 64) of women had HPV clearance (see Figure 4) [10,13,27,29,30]. We performed a subgroup analysis according to the grade of cervical dysplasia (see Figure 4). When CIN 2–3 was the diagnosis, the HPV clearance rate was 42% (CI: 0.29–0.56; *I*^2^: 49%); when CIN 1–3 was the diagnosis, the HPV clearance rate was 68% (CI: 0.48–0.84). Finally, when there was no CIN, only HPV positivity, the HPV clearance rate was 65% (CI: 0.44–0.83). However, we had only one study for each outcome. The subgroup analysis regarding the study types showed a 56% (CI: 0.28–0.80; *I*^2^: 59%) HPV clearance in the ITT-RCTs and 44% (CI: 0.17–0.75; *I*^2^: 73%) in the cohort studies (see Appendix A). Moreover, in the PP population of 100 patients, the HPV clearance rate was higher at 60% (CI: 0.35–0.84; *I*^2^: 57%) (see Appendix A). The subgroup analysis in the PP for CIN 2–3 showed a 54% HPV clearance (CI: 0.06–0.96; *I*^2^: 47%), and for CIN 1–3, the HPV clearance was 73% (CI: 0.52–0.88) (see Appendix A). The HPV tests were conducted on average 2.33 (SD: ±1.91) months after finishing Imiquimod treatment. 

The Imiquimod group had 196 patients, while the control arm had 180 patients. Imiquimod treatment did not result in better HPV clearance compared to that of the control group (RR: 1.29; CI: 0.52–3.21; *I*^2^: 80%) (see Figure 5). In the control group, the treatment differed between the studies. When the control group received conization, it was more effective than Imiquimod (RR: 0.67; CI: 0.46–0.99) [30]. However, when no intervention was implemented in the control group and HPV infection was persistent, Imiquimod was more effective (RR: 4.20; CI: 1.62–10.89) [11]. In another study, when there was only persistent HPV positivity and no cervical dysplasia, and the control arm received no intervention, Imiquimod was more effective (RR: 2.18; CI: 1.06–4.05) [29]. In one study, the control arm had surgical interventions (conization, cryotherapy, laser) [13]. We found that in this article, Imiquimod treatment did not result in better HPV clearance than the control group (RR: 1.19; CI: 0.79–1.79) [13]. When we examined the PP group, we concluded that Imiquimod treatment did not result in a higher HPV clearance rate compared to the control group (see Appendix A).

Among the 186 patients who received Imiquimod, we investigated HPV 16/18 clearance compared to the clearance of other high-risk HPV (HR-HPV) types [11,12,27,30]. Our findings indicate no significant difference between HPV 16/18 clearance and clearance of other HR-HPV types (RR: 0.89; CI: 0.58–1.37; *I*^2^: 0) (see Appendix A).

### 3.5. Adverse Events

In five studies, we were able to quantitatively synthesize the adverse events in patients who received Imiquimod, as they all used a very similar grading system (see Table 1) [10,11,13,28,30]. They graded side effects on a scale of one to five, where the grades ranged from mild to moderate, serious, life-threatening, and death (see Figure 6 and Appendix A).

The most common systemic side effects were flu-like symptoms and myalgia, while from the local side effects, vaginal pruritus was the most frequent.

Grade 3 side effects occurred 8 times, with two articles [10,28] reporting abdominal pain and two [28,30] reporting headache. The remaining four occurrences of grade 3 side effects were one flu-like symptom [11], one fever [30], one myalgia [30], and one vaginal inflammation [28].

### 3.6. Risk of Bias Assessment and GRADE

For randomized controlled studies, the ROB2 showed some concern for the risk of bias in five outcomes and a low risk of bias in two outcomes. In non-randomized clinical trials, the ROBINSON tool showed a moderate and serious risk of bias in two outcomes. The latter study was a retrospective cohort analysis that used a historical control group for comparison and had methodological concerns. When we analyzed the response rates according to the JBI critical appraisal checklist, we found that the most frequent issue was regarding sample size (see Appendix A).

Our summary of findings consisted of five outcomes where we included a control group. The quality of evidence was assessed as high for two outcomes, while it was deemed low for the remaining three outcomes (see Appendix A).

## 4. Discussion

We investigated the safety and efficacy of topical Imiquimod on cervical intraepithelial neoplasia and HPV clearance. After Imiquimod treatment in CIN 2–3 patients, the regression rate was 61%. Regarding efficacy, we analyzed the biopsies of CIN 2–3 patients and concluded that women who received Imiquimod treatment had a higher rate of CIN regression compared to those who did not receive Imiquimod. We found that conization is more effective than Imiquimod in treating CIN 2–3 patients, as there was a 38% increase in successful treatment in the conization arm.

Imiquimod treatment in HPV-positive women showed a 50% HPV clearance rate and was 60% for women who completed the treatment. Overall, Imiquimod treatment did not result in better HPV clearance compared to the control group’s treatment. When Imiquimod was compared to conization in one study, HPV clearance was higher in the conization group [30]. However, in another study comparing Imiquimod to placebo, HPV clearance was higher in the Imiquimod arm [11].

Regarding side effects, most side effects were mild, and hospitalization was not required in the majority of cases.

Increasing the dosage of Imiquimod did not result in a higher rate of CIN 2–3 regression. Regardless of the dose of Imiquimod, women who completed the treatment had a similar rate of dysplasia regression. In two studies, the remission rates were higher than in other studies [11,27]. In one study, the higher CIN 2/CIN 3 ratio could explain this. CIN 2 has a higher rate of spontaneous regression than CIN 3 and is considered a milder lesion [31,32]. The other study used cytological confirmation of CIN regression [27]. Although cytology is not a reproducible method of detecting cervical dysplasia [33], Imiquimod has been shown to be effective in reducing vulvar intraepithelial neoplasia and vaginal intraepithelial neoplasia [34,35]. The American College of Obstetricians and Gynecologists recognizes the off-label use of Imiquimod for vulvar intraepithelial neoplasia [36]. Our findings on Imiquimod in cervical dysplasia regression align with previous findings on other lower genital intraepithelial neoplasia [34,35]. Imiquimod is effective in reducing cervical CIN.

When comparing Imiquimod with the already existing surgical therapy for CIN 2–3, we found Imiquimod inferior. However, when the topical immunomodulator was used before conization, the positive margins of the resected tissue were lower than the average in the literature [10,37]. A logical explanation could be that Imiquimod reduced the lesion’s depth and width and made it more suitable for surgical excision [10]. For selected patients, Imiquimod can be a choice; for example, women with future pregnancy desires have a higher demand for conservative treatment [38], given that conization increases the risk of miscarriage and preterm birth by causing cervical incompetence [5]. The ASCCP guideline recommends diagnostic evaluation of the cervix after six months in case of positive surgical excision margins [4]. Implementing Imiquimod in positive margin cases could lower the need for additional surgical excision. This could be highly desirable in women considering future pregnancy, as repeated surgical intervention of the cervix increases the risk of preterm birth compared to one surgical excision of the cervix [39]. A recent study showed that patients who respond to Imiquimod treatment could be selected priorly with an immunohistochemical method, as the immune microenvironment predicts whether the patient will respond to Imiquimod treatment [40]. This could personalize Imiquimod treatment, as not all patients respond to it.

When examining the HPV clearance for HPV 16 and 18 and other HR-HPV types, we conclude that the clearance rate is not worse for HPV 16 and 18. This is interesting since HPV 16 and 18 are known to be more aggressive and accountable for 70% of all cervical cancer [41]. We investigated the best three HPV clearance rates in the studies. We found that in one article there were only persistent HPV infections without CIN lesions [29]. In another study, CIN 2/CIN 3 rate was the highest among all publications [11]. In the third study, the CIN 2/CIN 3 rate was high, and CIN 1 also occurred [13]. Moreover, in this study, the HPV tests were taken after 6 months of Imiquimod discontinuation, which is a long time considering the natural clearance of HPV [42]. Patients with CIN 2+ lesions are known to be, molecularly, a vastly heterogeneous population; with progression, the cellular changes are more extensive, and the spontaneous regression of CIN and HPV declines [43]. Our findings support this observation, as when more CIN 3 lesions occur, HPV clearance rates are lower with Imiquimod. Higher Imiquimod doses did not result in a higher HPV clearance rate. When Imiquimod was compared to the control group, we did not experience a better HPV clearance rate. However, it should be mentioned here that the control differed between studies: surgical excision of the HPV-infected area [30] and only expectant management with no intervention [11]. In the previous case, the surgical solution was comparable or more effective than Imiquimod; however, in the latter case, Imiquimod was more effective than expectant management.

Systemic and local side effects were frequent but mostly mild, and the symptoms could have been reduced with non-steroid anti-inflammatory drugs [11]. This variability can be attributed to the systemic side effects associated with Imiquimod, which other common infections or health-related conditions can influence. Additionally, local side effects caused by Imiquimod are commonly observed in general gynecological practice. This variability in side effects can help explain the differences observed among the studies.

Dropouts can happen for several reasons (long travel, financial reasons, dissatisfaction); however, it should be mentioned that in the two studies where the highest dose of Imiquimod was implemented, the two highest rates of dropout were also observed [12,28]. Severe side effects occurred just eight times, with high doses of Imiquimod used in seven of these cases. In two studies, besides the lower doses of Imiquimod used, the application was implemented by doctors. Accordingly, direct application could contribute to the low and mild side effects [10,27].

### 4.1. Strengths and Limitations

This meta-analysis is the first to synthesize the findings on Imiquimod use in cervical dysplasia and HPV-positive patients.

However, several limitations should be noted. Firstly, many studies had poor patient recruitment, and efforts were made to enroll more women. Secondly, the patients were mostly selected based on specific criteria, limiting the generalizability of the study’s implications to all cervical dysplasia patients. Thirdly, in some cases, the comparison was made with control groups that used different methods. Fourthly, many studies lacked longer follow-up intervals, raising questions about the durability of dysplasia remission. Fifthly, the endpoint and timing of different outcome measures were often inconsistent, which is problematic given the spontaneous tendency of these lesions to regress. Sixthly, the clinical and statistical heterogeneity was substantial in several cases.

### 4.2. Implications for Practice and Research

Our findings show Imiquimod is safe and effective in reducing CIN and facilitating HPV clearance. While Imiquimod is inferior to conization, it could still be considered for use in selected patients, particularly after positive margins of conization, to avoid subsequent surgical excision of the cervix.

Personalized treatment strategies hold promise for enhancing the therapeutic efficacy of Imiquimod, as the immune environment serves as a predictive factor for treatment success [40]. Further refinement of immunohistochemical methods to identify specific biomarkers could lead to even greater therapeutic responses, potentially revolutionizing the approach to CIN treatment. Investigating the microenvironment and molecular profiles in greater detail through focused studies has the potential to transform the utilization of Imiquimod in clinical practice. Additionally, exploring combined therapies involving Imiquimod and other agents, such as 5-Fluorouracil, may synergistically augment the reduction of CIN lesions [44]. This approach not only offers the possibility of reducing Imiquimod dosage but also of lowering potential side effects, thereby improving medication adherence. Determining the optimal dosage of Imiquimod is paramount, with further studies warranted to elucidate this aspect. Such investigations are motivated by the desire to minimize costs associated with Imiquimod, an expensive medication, while simultaneously mitigating side effects. Prior to the incorporation of Imiquimod into treatment guidelines, conducting cost–effectiveness analyses is imperative. Assessing the cost–benefit ratio compared to traditional treatments is essential for informing healthcare resource allocation decisions. A comprehensive understanding of these factors will be instrumental in optimizing the utilization of Imiquimod in the management of cervical dysplasia. Further interventional studies are needed in this field to better understand how Imiquimod can reduce the burden of cervical dysplasia. Particularly, investigations with extended follow-up periods are warranted, as current studies often lack prolonged monitoring. Consequently, elucidating the duration of Imiquimod’s effects remains an unanswered question that requires attention.

## 5. Conclusions

In conclusion, Imiquimod is not a substitute for cone biopsy; however, it can be a valuable treatment option for high-grade cervical dysplasia.

## Figures and Tables

**Figure 1 cancers-16-01610-f001:**
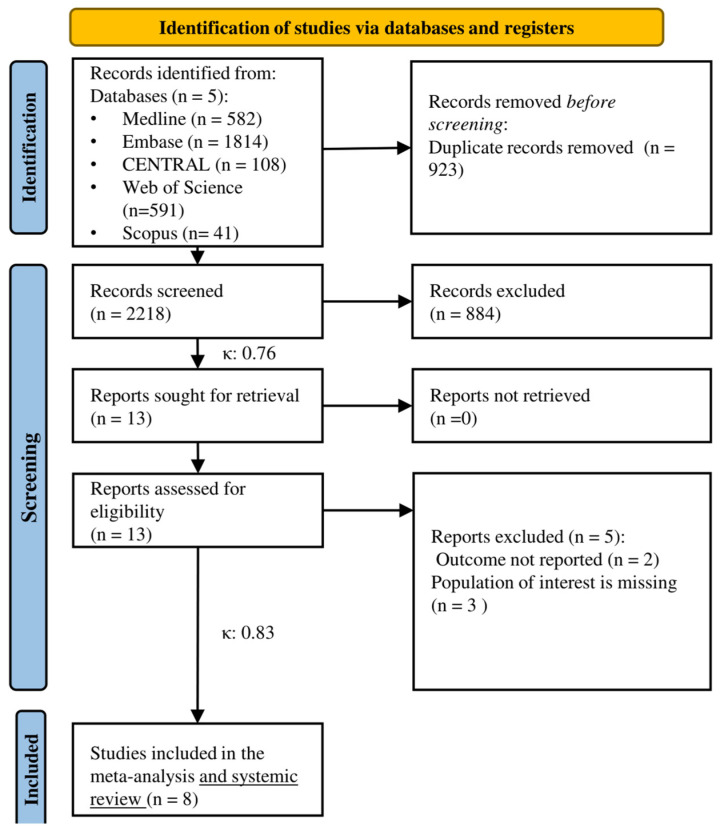
PRISMA flowchart of selection n: number of studies, κ: Cohen’s kappa coefficient.

**Figure 2 cancers-16-01610-f002:**
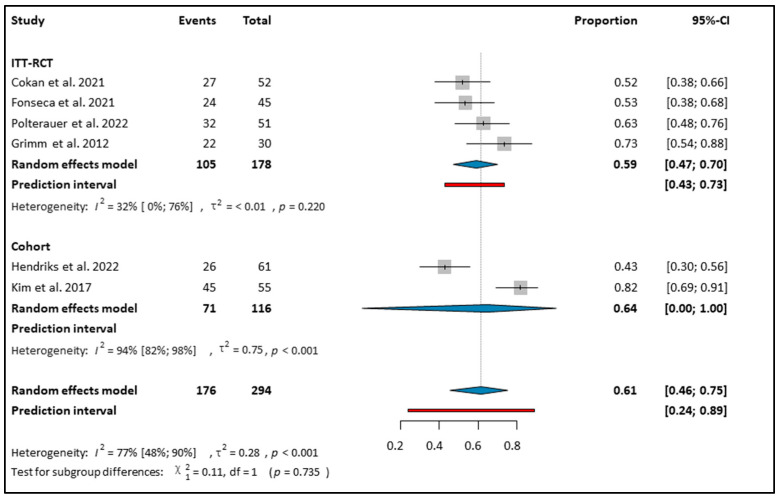
Forest plot of studies representing Imiquimod and CIN 2–3 regression based on study type [10,11,12,27,28,30]. ITT: intention to treat, RCT: randomized control trial. Effect Estimate: The effect estimate for each study is represented by a grey square, located along the x-axis. Confidence Interval (CI): A line extending from the effect estimate represents the confidence interval. This indicates the range within which the true effect size is likely to lie, with the point estimate positioned at the center of the bar. Overall Estimate: A summary effect estimate, represented by a blue diamond at the bottom of the plot, combines the results of all studies included in the meta-analysis. The center of the diamond represents the point estimate, and the width of the diamond represents the confidence interval around the summary estimate. *I*^2^ (I-squared): A measure of heterogeneity in meta-analysis, indicating the proportion of total variation across studies that is due to heterogeneity rather than chance. *p*-value (Probability value): The probability of obtaining test results at least as extreme as the observed results, indicating the significance of the heterogeneity test. χ^2^ (Chi-square): A test for subgroup differences, evaluating whether the observed differences between subgroups are statistically significant. df (degrees of freedom): The degrees of freedom, which represent the number of independent pieces of information used to estimate a parameter. τ^2^ (Tau-squared): The variance of true effects in a random-effects model, reflecting the variability of effect sizes across studies beyond sampling error.

**Figure 3 cancers-16-01610-f003:**
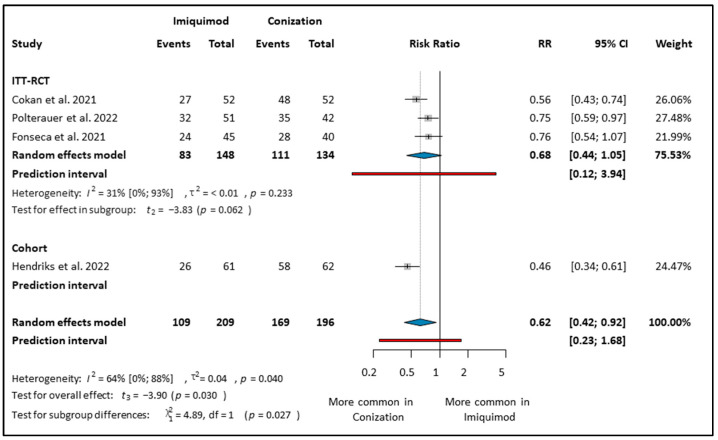
Forest plot of studies representing the Imiquimod group compared to conization on CIN 2–3 regression based on study type [10,12,28,30]. ITT: intention to treat, RCT: randomized control trial. Effect Estimate: The effect estimate for each study is represented by a grey square, located along the x-axis. Confidence Interval (CI): A line extending from the effect estimate represents the confidence interval. This indicates the range within which the true effect size is likely to lie, with the point estimate positioned at the center of the bar. Overall Estimate: A summary effect estimate, represented by a blue diamond at the bottom of the plot, combines the results of all studies included in the meta-analysis. The center of the diamond represents the point estimate, and the width of the diamond represents the confidence interval around the summary estimate. *I*^2^ (I-squared): A measure of heterogeneity in meta-analysis, indicating the proportion of total variation across studies that is due to heterogeneity rather than chance. *p*-value (Probability value): The probability of obtaining test results at least as extreme as the observed results, indicating the significance of the heterogeneity test. χ^2^ (Chi-square): A test for subgroup differences, evaluating whether the observed differences between subgroups are statistically significant. df (degrees of freedom): The degrees of freedom, which represent the number of independent pieces of information used to estimate a parameter. τ^2^ (Tau-squared): The variance of true effects in a random-effects model, reflecting the variability of effect sizes across studies beyond sampling error. Test for overall effect (*t*_3_): A statistical test used to assess whether the observed effect size is significantly different from zero, indicating the presence of an overall effect in the meta-analysis.

**Figure 4 cancers-16-01610-f004:**
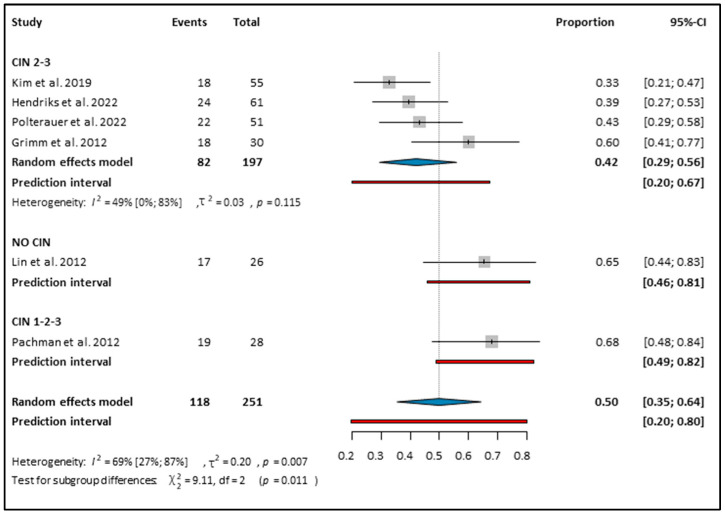
Forest plot of studies representing Imiquimod on HPV clearance based on the CIN status [11,12,13,27,29,30]. CIN: cervical intraepithelial neoplasia. Effect Estimate: The effect estimate for each study is represented by a grey square, located along the x-axis. Confidence Interval (CI): A line extending from the effect estimate represents the confidence interval. This indicates the range within which the true effect size is likely to lie, with the point estimate positioned at the center of the bar. Overall Estimate: A summary effect estimate, represented by a blue diamond at the bottom of the plot, combines the results of all studies included in the meta-analysis. The center of the diamond represents the point estimate, and the width of the diamond represents the confidence interval around the summary estimate. *I*^2^ (I-squared): A measure of heterogeneity in meta-analysis, indicating the proportion of total variation across studies that is due to heterogeneity rather than chance. *p*-value (Probability value): The probability of obtaining test results at least as extreme as the observed results, indicating the significance of the heterogeneity test. χ^2^ (Chi-square): A test for subgroup differences, evaluating whether the observed differences between subgroups are statistically significant. df (degrees of freedom): The degrees of freedom, which represent the number of independent pieces of information used to estimate a parameter. τ^2^ (Tau-squared): The variance of true effects in a random-effects model, reflecting the variability of effect sizes across studies beyond sampling error.

**Figure 5 cancers-16-01610-f005:**
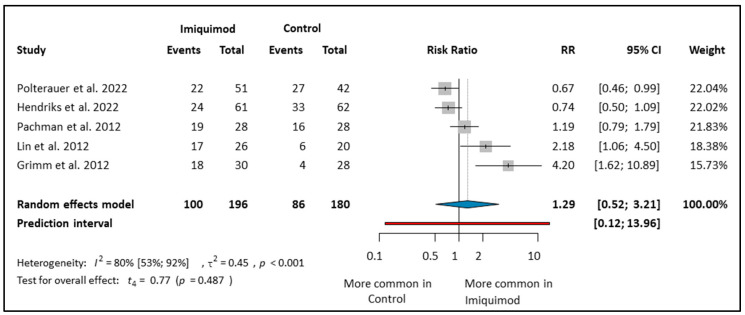
Forest plot of studies representing the Imiquimod group compared to control on HPV clearance [11,12,13,29,30]. Effect Estimate: The effect estimate for each study is represented by a grey square, located along the x-axis. Confidence Interval (CI): A line extending from the effect estimate represents the confidence interval. This indicates the range within which the true effect size is likely to lie, with the point estimate positioned at the center of the bar. Overall Estimate: A summary effect estimate, represented by a blue diamond at the bottom of the plot, combines the results of all studies included in the meta-analysis. The center of the diamond represents the point estimate, and the width of the diamond represents the confidence interval around the summary estimate. *I*^2^ (I-squared): A measure of heterogeneity in meta-analysis, indicating the proportion of total variation across studies that is due to heterogeneity rather than chance. *p*-value (Probability value): The probability of obtaining test results at least as extreme as the observed results, indicating the significance of the heterogeneity test. τ^2^ (Tau-squared): The variance of true effects in a random-effects model, reflecting the variability of effect sizes across studies beyond sampling error. Test for overall effect (*t*_4_): A statistical test used to assess whether the observed effect size is significantly different from zero, indicating the presence of an overall effect in the meta-analysis.

**Figure 6 cancers-16-01610-f006:**
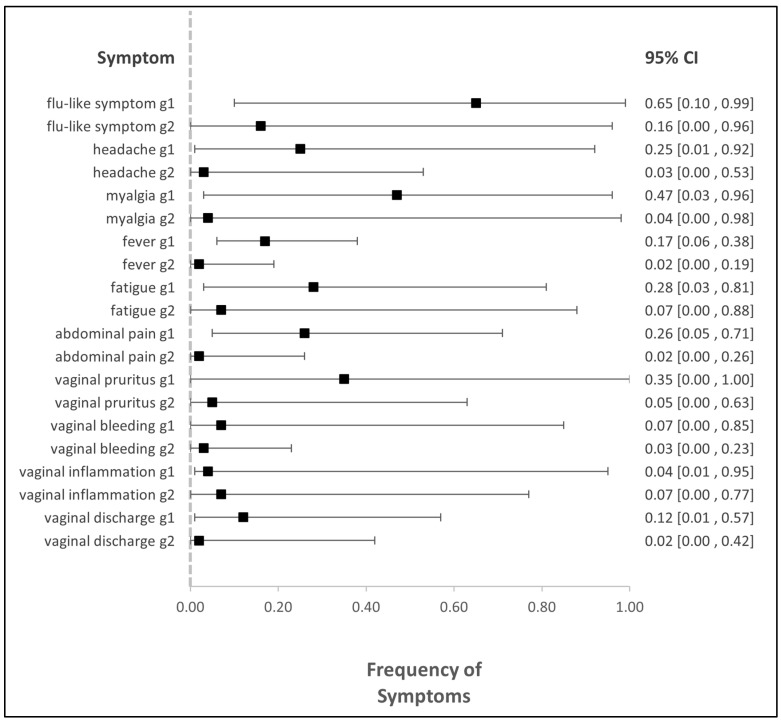
Forest plot of studies representing the frequency of all symptoms. g1: grade1, g2: grade2, CI: confidence interval.

**Table 1 cancers-16-01610-t001:** Basic characteristics of included studies.

Author, Years	Study Type	Region	Follow Up Time (Months)	Number of Patients in Intervention	Age (Mean) Intervention, SD	Number of Patients in Control	Age (Mean) Control, SD	CIN ^B^	CIN2/CIN3 Ratio	HPV ^D^ Type	Dose of Imiquimod/Patient	Intervention of Control Group	Adverse Event Reporting	Dropout of Patients
Grimm et al., 2012 [11]	RCT ^A^	Austria	5	30	29.2 ± 6.1	29	31.8 ± 7.8	CIN 2–3	1.73	HPV 16/18, other HR ^E^ HPV	243.75 mg	observation	CTCAE ^F^ 3.0	6.70%
Hendriks et al., 2022 [12]	Non-randomized interventional	The Netherlands	6	61	33.3 ± 9.1	62	35.2 ± 7	CIN 2–3	0.69	HPV 16/18, other HR HPV	300 mg	conization	VAS ^G^	22.90%
Cokan et al., 2021 [28]	RCT	Slovenia	6	52	28.3 ± 4.2	52	26 ± 4.6	CIN 2–3	0.79	NA	600 mg	conization	CTCAE 5.0	17.30%
Lin et al., 2012 [29]	Retrospective cohort analysis	Taiwan	33.4	72	51.75 ^B^	20	50 ^B^	NA ^C^	NA	persistent HR-HPV	150 mg	observation	NA	NA
Fonseca et al., 2021 [10]	RCT	Brazil	24	45	32 ^B^	45	36 ^B^	CIN 2–3	0.4	NA	150 mg	observation	CTCAE 4.0	15.60%
Pachman et al., 2012 [13]	RCT	USA	37.2	28	30 ± 8.9	28	29 ± 9.7	CIN 1–2–3	1.42	HR-HPV	12.5 mg	conization, laser, cryotherapy	CTCH ^H^ 2.0	7.14%
Polterauer et al., 2022 [30]	RCT	Austria	24	51	31.4 ^B^	42	30.1 ^B^	CIN 2–3	0.28	HPV 16/18, other HR HPV	243.75 mg	conization	CTCAE 3.0	9.80%
Kim et al., 2019 [27]	retrospective cohort analysis	Republic of Korea	13.4	55	30 ^B^	NA	NA	CIN 2–3	0.74	HPV 16/18, other HR HPV	100 mg	NA	NA	1.80%

^A^ randomised control trial; ^B^ median cervical intraepithelial neoplasia; ^C^ Not applicable; ^D^ Human papillomavirus; ^E^ High-risk; ^F^ Common Terminology Criteria for Adverse Events; ^G^ Visual Analog Scale; ^H^ Common Toxicity Criteria.

## Data Availability

The dataset utilized in this meta-analysis is available in the full-text articles included in this systematic review.

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
