# Peer review of "Imiquimod Is Effective in Reducing Cervical Intraepithelial Neoplasia: A Systematic Review and Meta-Analysis"

_cancers, 2024, doi:10.3390/cancers16081610_

Round 1

Reviewer 1 Report

Comments and Suggestions for Authors

The study by Hamar et al. discusses about the potential of imiquimod for cervical intraepithelial neoplasia.

Some suggestions to improve this manuscript are:

1. The font size of the figures should be enlarged for better view.

2. The figure legends should be significantly elaborated with more details.

3. Figure should be represented in a better way.

4. In the discussion section, more details should be included in terms of future directions.

5. A schematic diagram can be included summarizing the findings and future directions.

Author Response

Dear Reviewer,

Many thanks for your review. I’ve greatly appreciated your comments which have significantly improved our manuscript. Please find our point-by-point response to your comments below. I hope that our revised manuscript will meet your expectations and you will find it suitable for publication. Please let me know if any further changes are required.

  1. The font size of the figures should be enlarged for better view.

Repsponse: The front size of the figures have been enlarged

  1. The figure legends should be significantly elaborated with more details.

Response: The figure legends are revised, and updated with a more precise description

  1. Figure should be represented in a better way.

Response: Figures presentation has been changed to gain a better visualisation

  1. In the discussion section, more details should be included in terms of future directions.

Please see this paragraph under “Implication for Practice and Research”

“Personalized treatment strategies hold promise for enhancing the therapeutic efficacy of Imiquimod, as the immune environment serves as a predictive factor for treatment success. Further refinement of immunohistochemical methods to identify specific biomarkers could lead to even greater therapeutic responses, potentially revolutionizing the approach to CIN treatment. Investigating the microenvironment and molecular profiles in greater detail through focused studies has the potential to transform the utilization of Imiquimod in clinical practice. Additionally, exploring combined therapies involving Imiquimod and other agents, such as 5-Fluorouracil, may synergistically augment the reduction of CIN lesions. This approach not only offers the possibility of reducing Imiquimod dosage but also lowering potential side effects, thereby improving medication adherence. Determining the optimal dosage of Imiquimod is paramount, with further studies warranted to elucidate this aspect. Such investigations are motivated by the desire to minimize costs associated with Imiquimod, an expensive medication, while simultaneously mitigating side effects. Prior to the incorporation of Imiquimod into treatment guidelines, conducting cost-effectiveness analyses is imperative. Assessing the cost-benefit ratio compared to traditional treatments is essential for informing healthcare resource allocation decisions. A comprehensive understanding of these factors will be instrumental in optimizing the utilization of Imiquimod in the management of cervical dysplasia.

Further interventional studies are needed in this field to better understand how Imiquimod can reduce the burden of cervical dysplasia. Particularly, investigations with extended follow-up periods are warranted, as current studies often lack prolonged monitoring. Consequently, elucidating the duration of Imiquimod's effects remains an unanswered question that requires attention.”

  1. A schematic diagram can be included summarizing the findings and future directions.

Response: We performed a graphical abstract that summarizes the main findings and future directions. If you prefer another diagram, please don’t hesitate to request it.

Reviewer 2 Report

Comments and Suggestions for Authors

Summary:

This papers analysis the effectiveness of the immune modifier Imiquimod in treating CIN 2/3 in a systematic review and meta-analysis of 8 elective papers. 398 patients received Imiquimod. The regression rate of CIN 2/3 patients who received the immune response modifier was compared to patients treated by conization, placebo or only observational treatment. So far no meta-analysis on this topic has been conducted and published. The strength of the paper is the systematic review and extensive search of the literature and papers using Imiquimod in the treatment of CIN 2/3 compared with the standard treatment conization or observation. Furthermore the (known) side effects are graded and analysed.

General concept comments:

This review is very informative and relevant for all clinicians treating patients with CIN 2/3, especially young women who still have pregnancy wish and want to be treated without surgical intervention. Surgery of CIN 2/3 bears the (low)  risk of preterm birth and cervical insufficiency, therefore there is a need of evaluating a conservative treatment option like Imiquimod. The review fills a gap in knowledge of a conservative treatment option of patients with CIN 2/3 . The meta-analysis is extensively performed, complete, the topic relevant and the presentation of the data and results comprehensive and understandable.

The structure of the paper is clear, the number of references adequate and the most relevant papers includes. No self citations. The results and figures and tables are appropriate and very detailed and sufficient to show the results of the statistic analysis.

The conclusion is adequate due to the results of the systematic review and analysis of the relevant studies.

Minor comments:

there is one reference which should be included in the analysis:

J.A. Lieb, A. Mondal, L. Lieb, TN Fehm, M. Hampl (2022) Pregnancy outcome and risk of recurrence after tissue preserving loop excision procedure (LEEP). Arch Gynecol Obstet , September, 2022 should be added at line 55 : .....also data concerning pregnancy outcome are controversial (5, and new)

Author Response

Dear Reviewer,

Thank you for your thorough review. I found your comments interesting and valuable. I hope the required point has been dealt with and you find it worthy of publication. Please let me know if any further changes are needed.

We included the mentioned article in the reference.

Round 2

Reviewer 1 Report

Comments and Suggestions for Authors

The revised manuscript looks better.